# A Study on the Structural Relationships between COVID-19 Coping Strategies, Positive Expectations, and the Behavioral Intentions of Various Tourism-Related Behaviors

**DOI:** 10.3390/ijerph20021424

**Published:** 2023-01-12

**Authors:** Mengen Zhang, HakJun Song

**Affiliations:** College of Tourism and Fashion, Pai Chai University, Daejeon 35345, Republic of Korea

**Keywords:** coping strategies, attitude, positive anticipated emotion, positive expectation of COVID-19, behavioral intention, COVID-19 quarantine policy, indoor leisure, outdoor leisure, intra-bound tourism, smart tourism

## Abstract

The purpose of this study was to investigate the impact of coping strategies, attitudes, and positive anticipated emotions on the positive expectations and behavioral intentions of Korean tourists during the COVID-19 pandemic. An integrated model was proposed and tested, and the results indicate that effective coping strategies, attitudes, and positive anticipated emotions have a positive effect on the positive expectations of tourism during the pandemic, which in turn positively influences behavioral intentions. Practical suggestions were also provided based on the findings. This research has implications for understanding the ways in which individuals cope with and adapt to travel during times of crisis, and for identifying strategies that may facilitate positive expectations and behavioral intentions in the tourism industry.

## 1. Introduction

The tourism industry is susceptible to various natural and human-made threats that can disrupt and damage tourism destinations. Natural disasters such as hurricanes, earthquakes, and volcanic eruptions can cause destruction to infrastructure and natural attractions, leading to a decline in tourism. For example, Cancun, a popular tourist destination in Mexico, was severely damaged by hurricanes Gilberto in 1988 and Wilma in 2005, which had significant impacts on the tourism industry in the region. In addition to natural disasters, environmental crises such as the bladder-wrack crisis that affected the entire Mexican Caribbean in 2019 can also negatively impact tourism. This crisis, caused by an invasive seaweed species that impacted the region’s beaches and coral reefs, led to a decline in tourism in the affected area. These types of threats can have significant economic and social impacts on tourism destinations and the people who depend on tourism for their livelihoods [1,2,3]. Wu and Hayashi have divided these disasters into three types: geological disasters, extreme weather events, and other disasters (e.g., terrorist attacks, infectious diseases, and economic crises) [4]. These episodes have put tourism activity at risk in different touristic territories on a global scale. Differently from the recent epidemic outbreaks such as SARS, Ebola, and H1N1 (Influenza A virus subtype), Coronavirus SARS-CoV-2 disease 2019 (hereafter COVID-19) remains the world’s deadliest epidemic outbreak that comes along with a systemic global healthcare crisis, financial crisis, and economic downturn known as COVID-19 recession [5]. As of 20 April 2022, more than 504.4 million confirmed COVID-19 cases and over 6.2 million related deaths had been reported to WHO. Approximately, 1.2 million new cases were being reported every 24 h in early 2022, but this has now decreased to about 400,000 during April 2022 [6]. The COVID-19 pandemic has restricted travel and led to a sharp decline in the number of tourists, with serious implications for travel, tourism, and hospitality around the world, including the Republic of Korea (hereafter “Korea”) the subject of the current study [7,8].

In order to continue operating the tourism industry during the COVID-19 pandemic, various stakeholders must carefully implement strategies and policies as coping actions. South Korea has developed and implemented a range of coping strategies in response to the various risks facing the tourism industry, including the rapid implementation of existing response plans, innovative measures for unexpected risks, and adaptive responses as the pandemic evolves [9]. It is important for tourism managers to develop and implement strategies to revitalize the tourism industry [10], and for governments to provide appropriate policies to support this. Both the government and the travel business community in South Korea have tried to convey a sense of calm and containment in response to the outbreak. To ensure the safety of tourism activities in destinations, tourists should also implement personal COVID-19 coping strategies. Many countries have implemented measures such as lockdowns and confinement in order to control the spread of the pandemic, although the extent and duration of these measures vary.

There is a significant amount of literature on the topic of the COVID-19 pandemic, but relatively few studies have focused on the impact of coping strategies on behavioral intentions in the tourism industry. The COVID-19 pandemic has not only disrupted ongoing tourism activities, but it has also affected the willingness of individuals to engage in future tourism-related activities. For example, a study by Wachyuni and Kusumaningrum (2020) analyzed the post-pandemic travel intentions of tourists and found that there is hope and optimism that the tourism industry will recover in the near future. However, further research is needed to fully understand the relationship between coping strategies and behavioral intentions in the context of the COVID-19 pandemic, particularly as it relates to the tourism industry [11]. A study on Serbian tourists showed that the risk perception of an infectious disease had a negative effect on travel intentions during the epidemic period [12]. Bae and Chang (2021) highlighted ‘untact’ tourism as a health-protective behavior stemming from individuals’ perceptions of the COVID-19 risk [13].

The study establishes a theoretically integrated framework for exploring relationships between coping strategies, expectations and behavioral intentions in the tourism context. The findings of this study would contribute to a better understanding of tourists’ COVID-19 expectations towards travel intent and to develop effective communications and policies to control the level of perceptions of COVID-19 and encourage travel.

## 2. Literature Review

### 2.1. Coping Strategies

The impact of COVID-19 on the economy and society is enormous, and the tourism sector is no exception. One study explored the literature and reported the coping strategies related to the COVID-19 pandemic of the tourism and hospitality industries and provided recommendations for the future of the tourism industry [14]. This study suggests that various stakeholders’ strategies for coping with COVID-19 are necessary as a multi-faceted effort to maintain, improve, and develop the tourism industry despite the COVID-19 situation. The thoughts and behaviors that people use to manage the stressful internal and external demands of situations are defined as coping strategies [15]. Coping is usually referred to a behavioral reaction to aversive situations [16,17,18]. Most commonly, coping is described from two main perspectives: changing the relationship with the environment with coping actions (problem-focused coping) or changing the interpretation of the environment (emotion-focused coping) [19]. Specifically, the response strategies for COVID-19 can be subdivided into individual, tourism companies, and government response strategies because the stakeholders’ responses to a situation can be different from each other.

Several studies have been conducted on individual coping strategies for infectious diseases. Specifically, Rizzo and Atti (2008) introduced pharmaceutical and non-pharmaceutical interventions as methods of suppressing the influenza virus [20]. A study stated that non-pharmaceutical interventions which centered on hygiene rules such as social distancing and hand washing effectively contributed to the prevention of COVID-19 infection [21]. Such individual coping strategies can be applied not only to general social activities, but also to tourism. The responses of tourists and the demands of tourism can only occur when the tourism supply from tourism businesses is stabilized. This indicates that the strategies of tourism businesses for coping with COVID-19 should be considered as important [22]. Many tourism-related businesses around the world are providing customers with sufficient information about COVID-19 along with the regular quarantine procedures. In addition, they are promoting various quarantine activities, such as checking body temperature upon entry, preparing disinfectants, and requesting quarantine-related calls. The preparation of response procedure manuals, the establishment of a hotline with related organizations, and employee training were also being actively carried out within tourism businesses during COVID-19. Dayour et al. (2020) explored Ghana’s SMHTO (Small and Medium Hotels and Tour Operators) coping strategies against COVID-19, and they presented Ghana’s coping strategies for COVID-19 and the long-term and short-term post-recovery strategies [23].

In addition, the coping strategies of individuals and tourism companies as well as the national coping strategy (i.e., policy) for COVID-19 can play an important role in the tourism industry so that successful operation in a tourism crisis occurs. A government’s COVID-19 Quarantine Policy generally includes various measures such as social distancing, self-isolation, and reduced travel [24]. Moreover, in response to the international tourism crisis, governments need to focus on boosting domestic tourism [25]. For example, in Italy there is the “bonus vacanze”—a voucher of up to 500 euros to stimulate domestic tourism. Allaberganov and Mohammed (2021) explored the tourism policy of the government of Uzbekistan to bring back the tourists and sustain the tourism and hospitality sector during the COVID-19 pandemic [26]. Ogueji et al. (2021) surveyed individual coping strategies during the COVID-19 pandemic using 50 U.K. residents and suggested that the U.K. employed aggressive and maladaptive coping strategies during the COVID-19 pandemic [27]. The COVID-19 coping strategies of individuals, tourism companies, and countries are very important, and many tourism activities are being focused accordingly; however, there is a lack of research exploring this as an important topic in tourism research.

### 2.2. Attitude and Positive Anticipated Emotions

Attitude refers to an individual’s positive or negative assessment of a particular phenomenon or action [28]. Considering that an attitude lasts for a long time once it is formed, it often acts as a meaningful predictor of an individual’s behavioral intention [29]. The COVID-19 public health crisis has not only affected the development, process, and sustainability of the tourism industry, but has also influenced tourists’ attitudes, needs, and behaviors [30,31]. For example, Sigala (2020) showed that tourists’ experiences during COVID-19 will have a significant influence on their travel attitudes, travel risk perceptions, intentions, and future behaviors [32,33]. Kamata (2022) assessed the reactions of Japanese residents towards tourism during and after the COVID-19 pandemic [34]. This study’s implications helped the destination managers communicate with residents to promote tourism sustainably. Zhang, Ruan, and Liu (2022) utilized a big data sentiment analysis to gain insight into tourists’ attitudes towards virtual tourism in crisis situations [35].

People may have forward-looking emotions toward uncertain future behaviors [36]. In particular, expecting psychological benefits from performing a specific behavior causes positive anticipated emotions, whereas expecting psychological damage from failing to perform the behavior leads to negative anticipated emotions [37]. This indicates that anticipated affective pre-responses to the performance of a behavior can be important determinants of behavioral intention [38,39,40,41]. Many studies have used anticipated emotions to explore tourists’ behavioral intentions during the pandemic. Das and Tiwari (2021) revealed that attitudes, subjective norm, perceived behavioral control, and positive anticipated emotions, all positively influenced travelers’ intentions, and negative anticipated emotions negatively influenced travelers’ intentions, relating to their desire to travel during COVID-19 [42]. Foroudi, Tabaghdehi, and Marvi (2021) examined customers’ perceptions of the shock of COVID-19 on their beliefs, and how their beliefs could influence their anticipated emotions and affect their future desire toward visiting restaurants [43].

### 2.3. Expectations

According to Higgs, Polonsky, and Hollick, expectations refer to the predictions or anticipations that consumers have about the future performance or outcomes of products or services. These expectations can be influenced by a variety of factors, including past experiences with the product or service, advertising and marketing efforts, and the reputation of the company or brand. Expectations play a crucial role in shaping consumer behavior and decision-making, as they can affect whether a consumer chooses to purchase a product or service, and their level of satisfaction with it. It is important for companies and organizations to manage and align their products and services with customer expectations in order to ensure customer satisfaction and loyalty [44]. Feather (1988) argued that what a person does in a situation is related to the subjective value of the expectations held by that person and the consequences that can occur following the action taken. In this regard, it seems that people have always believed that expectations are important for explaining individual behavior, especially their economic behavior [45]. In the marketing literature, expectations have been regarded as the benchmark used by consumers to determine product or service satisfaction or evaluate performance [46].

There are two kinds of expectations: expectations of what “will” happen and expectations of what “should” happen. The former refers to predictions based on past experiences or frequent events, while the latter refers to what consumers hope will happen based on their ideal or potential needs and the information they have received from numerous sources, including word-of-mouth [47]. In the tourism industry, tourists usually have initial expectations for the product or service before using the product or service. These expectations are formed through the information in advertisements and the reputation of other consumers in the past [48]. Del Bosque et al. (2006) showed that expectations were formed by experience, tourists’ previous satisfaction with services, communication between service providers (such as promises), and tourists’ perceptions of services [49]. In this regard, tourists’ expectations of COVID-19 could greatly affect their behavioral intentions. Park and Ahn (2022) examined government employees’ experiences and expectations of socioeconomic hardship during the COVID-19 pandemic [50]. The study found that women and racial minorities have had worse experiences and expectations of pandemic difficulties than men and non-Hispanic whites. The findings also reveal a clear gender gap and racial disparities in the experience and expectations of pandemic hardships. De Simone et al. (2022) investigated the relationship between citizens’ economic expectations and citizens’ satisfaction with democracy amid the pandemic in Europe. They showed that the COVID-19 pandemic has profoundly affected citizens’ economic expectations [51].

### 2.4. Behavioral Intentions of Various Tourism-Related Behaviors

Intention usually refers to the determination to take a specific action in the future, representing the probability of putting one’s beliefs into practice, and behavioral intention is often used as a decision to act [52]. Behavioral intention is the individual’s belief and will to represent the consumer as a specific future behavior after forming an attitude toward an object [53].

The behavioral intentions of tourism can be achieved in various ways during the COVID-19 pandemic. Firstly, the behavioral intention of intra-bound tourism can be significantly increased because it can be perceived that this tourism is relatively safer than traveling aboard. For example, during the pandemic, border closures, bans on public gatherings, and restrictions on movement between countries have negatively impacted the demand for international travel. Therefore, it is considered that there were many cases of tourists, who usually preferred overseas travel, deciding that their tourist destination was not overseas but domestic during the COVID-19 pandemic [54,55,56]. Secondly, many studies have proposed that smart tourism has played a crucial role in reducing the outbreak of epidemics and there is a growing demand for smart tourism by tourists [57,58,59,60,61].

Meanwhile, the continued spread of COVID-19 is thought to have affected leisure activities as well as tourism. Participation in leisure activities can have a positive impact on physical and mental health, and it is believed that the COVID-19 outbreak has caused significant changes in leisure activities [62,63,64,65,66]. Leisure activities can be divided into indoor leisure and outdoor leisure [67]. Specifically, outdoor leisure activities have increased more than indoor leisure activities due to the transmission of the fear of COVID-19 in indoor environments. Outdoor leisure has been shown to help people recover physically and mentally compared to indoor leisure [68]. Jackson et al. (2021) suggested that participation in outdoor activities during the pandemic improved mental health and well-being in adolescents [69].

However, it is considered that the indoor leisure activities of online shopping, online movie viewing, and indoor personal fitness have increased. In Taiwan, for example, it has been reported that people like to share their indoor leisure activities through social media. Therefore, tour operators have launched virtual online tourism, to continue to interact with the market and maintain good relationships with customers via social media [70]. People’s outdoor leisure activities have not reduced, but have slightly modified or even increased (e.g., camping) [71]. The results of a study of South Korean residents during the pandemic showed that they proactively overcame recreational restrictions [72].

### 2.5. Theoretical Framework

The framework for the present study explored the relationships between coping strategies for COVID-19, attitudes, positive anticipated emotions and Positive Expectations of COVID-19 (PEC).

Research has shown that there are expectations that the general assumptions about one’s life and the world at large are related to a reaction to sudden and unpredictable danger [73]. Zare et al. (2021) investigated the Iranian government’s response to the COVID-19 pandemic and they revealed that people expect the government and other responsible institutions to minimize the burden of this pandemic through adopting effective policies [74]. Bhatt et al. (2020) explored the perceptions of people towards COVID-19 and their experiences during the pandemic in Nepal. The respondents expected that the use of masks, sanitizers, handwashing and effective lockdowns would help to prevent the disease [75]. Gao et al. (2020) explored the factors influencing beliefs about COVID-19 in outpatients with mood disorders [76]. These researchers found that under the measures taken for epidemic prevention and coping strategies, the outpatients’ future epidemic prospects were inclined to positive expectations, but that their attitudes were more negative than the general public. Verhage et al. (2021) explored how older adults were coping and what they expected from COVID-19 [77]. Chua et al. (2020) proposed that in the interests of corporate social responsibility companies intervene in order to reduce the misfortune caused by catastrophes through involvement with the community, the market, and the supply chain players [78]. Similarly, it was confirmed that travel companies’ approaches to coping with COVID-19 can also affect tourists’ positive COVID-19 expectations [79,80,81].

Hsu et.al. (2010) identified the interrelationships between expectations, motivation, and attitudes using data collected from potentially outbound Chinese tourists to Hong Kong [29]. Drawing upon an expectations, motivation, and attitudes (EMA) model, a study examined the correlation between traveler expectations, attitudes, and motivation for visiting an outbound destination [82]. Carrera, Caballero, and Munoz (2012) showed the influence of emotions in the prediction of expectations [83].

Therefore, this paper proposes the hypothesis that effective coping strategies, attitudes, and positive anticipated emotions are conducive to the formation of positive expectations:

**Hypothesis** **1 (H1).**
*Personal COVID-19 Quarantine Policy (PQP) has a positive influence on PEC;*


**Hypothesis** **2 (H2).**
*Company’s COVID-19 Quarantine Policy (CQP) has a positive influence on PEC;*


**Hypothesis** **3 (H3).**
*Policy (PLY) has a positive influence on PEC;*


**Hypothesis** **4 (H4).**
*Attitude (ATT) has a positive influence on PEC;*


**Hypothesis** **5 (H5).**
*Positive Anticipated Emotion (PAE) has a positive influence on PEC.*


The relationship between the Positive Expectations of COVID-19 (PEC) and Behavioral Intention (BI) are discussed below.

Bhattacherjee (2001) proposed the Expectation–Confirmation Model (ECM) to understand the relationship between expectations, satisfaction, inconsistency, attitudes, and intentions in the context of tourism and hospitality. This model is based on the Expectation–Confirmation Theory (ECT) and the Technology Acceptance Model (TAM). Joo and Shin (2017) conducted a study to investigate the structural relationships between students’ expectations, perceived enjoyment, perceived usefulness, satisfaction, and continuance intention to use digital textbooks in middle school, using Bhattacherjee’s (2001) ECM as a theoretical framework. Specifically, the researchers wanted to understand how these factors influenced students’ decisions to continue using digital textbooks in the future [84,85]. This study confirmed that: (a) the more expectations of satisfaction that are placed on digital textbooks, the higher the perception of enjoyment and usefulness, (b) perceived enjoyment and perceived usefulness indirectly affect the continuance intention to use digital textbooks by mediating satisfaction, and (c) perceived usefulness and satisfaction directly influence the continuance intention to use, while perceived enjoyment did not significantly affect the continuance intention to use. Although the epidemic has caused people to participate more in indoor activities, when people have positive expectations of COVID-19, they also participate in outdoor recreational leisure activities [86,87,88]. Arbulú, Razumova, Rey-Maquieira, and Sastre (2021) assessed the ability of Spain’s domestic tourism industry to mitigate the country’s tourism crisis in the event of a complete or significant loss of international demand [54]. In addition, it seems that the COVID-19 has also changed the way people travel so that they are transitioning from traditional travel to smart tourism [89,90,91]. In the tourism industry, many previous studies have also confirmed the relationship between expectations and behavioral intentions [92,93,94]. Therefore, this paper divides behavioral intentions into four areas (Indoor Leisure, Outdoor Leisure, Intra-bound Tourism, and Smart Tourism) to analyze the relationships between people’s positive expectations of COVID-19 and their behavioral intentions:

**Hypothesis** **6 (H6).**
*PEC has a positive influence on Indoor Leisure (IDL);*


**Hypothesis** **7 (H7).**
*PEC has a positive influence on Outdoor Leisure (OTL);*


**Hypothesis** **8 (H8).**
*PEC has a positive influence on Intra-bound Tourism (IBT);*


**Hypothesis** **9 (H9).**
*PEC has a positive influence on Smart Tourism (SMT).*


Based on the research covered in the literature review and the proposed hypotheses developed above, the research model was developed and is illustrated in Figure 1 below.

## 3. Methodology

The measures of the constructs in the present study were employed from the extant literature [24,48,51,92,93,94,95,96,97]. All items were measured using a five-point Likert-type scale, ranging from strongly disagree (1) to strongly agree (5). Specifically, the coping strategies were operationalized by 12 items (4 items of Personal COVID-19 Quarantine Policy (PQP), 4 items of Company’s COVID-19 Quarantine Policy (CQP), 4 items of Policy (PLY)). A total of 3 items were utilized in order to measure awareness of the Attitude (ATT). In addition, we used 3 items for the Positive Anticipated Emotion (PAE). Positive Expectation of COVID-19 (PEC) was operationalized by 4 items. Lasty, behavioral intention was operationalized by 14 items (3 items of Indoor Leisure, 3 items of Outdoor Leisure, 3 items of Intra-bound Tourism, and 5 items of Smart Tourism).

With the development of the Internet some researchers in tourism use online surveys to reach broader populations of interest more efficiently [98,99]. This study used data collected by the Korean Internet survey firm that utilizes a nationwide panel of online respondents from which representative samples are selected. The research was mainly aimed at Korean citizens. The respondents were able to participate in the online survey by clicking the Participate-in-Survey button. They were selected as the most suitable participants through their responses to a sample questionnaire, and if they completed the questionnaire too quickly, they would be rejected during the survey. The survey was conducted from 20 October to 25 October 2021. An explanation of the purpose of the research and the response time was provided as well as compensation.

The Internet survey firm distributed questionnaires to 500 tourists chosen randomly. The statistical packages of Rstudio were employed to analyze the collected data using structural equation modeling (SEM). SEM tested how the collected data were explained by the proposed model [100] and used a two-step hybrid method which specified a measurement model in the confirmatory factor analysis and tested the structural model developed from the measurement model [101,102,103]. Through this procedure, a total of 375 completed and usable cases were gathered from the participants, their ages ranging from 10 to 60 years.

## 4. Results

### 4.1. Descriptive Statistics

Table 1 presents the demographic characteristics of the respondents. The proportion of male respondents (51.73%) was higher than the proportion of female respondents (48.27%). The age groups of the respondents were 20–29 years (17.60%), 30–39 years (24.27%), 40–49 years (30.40%), and 50–59 years (27.73%). More than half (58.13%) of the respondents were university students. The most common monthly income level of respondents was KRW 2 to 2.9 million (27.47%), followed by income levels of KRW 3 to 3.9 million (18.13%), and less than KRW 1 million (13.33) (the exchange rate at the time of the survey was KRW 1134 to USD 1). Married respondents (58.67%) outnumbered single respondents (39.73%). The most common employment for respondents was office workers (41.33%), followed by experts/professionals and housewives (12.80%).

### 4.2. Measurement Model

SEM research usually employs the maximum likelihood estimation method under the assumption that collected data have a multivariate normal distribution [104]. However, most data in the social sciences appear not to have multivariate normal distributions [105]. This study employed Mardia’s standardized coefficient to confirm whether or not the data in the study violate the assumption of multivariate normality [104]. In this study, the data appear to be multivariate non-normally distributed data, as Mardia’s standardized coefficient for the measurement model (71.445) is greater than the criterion of 5 [104]. When the collected data have a multivariate non-normal distribution, the robust maximum likelihood method based on the Satorra–Bentler (S–B) χ^2^ can provide more stable and robust standardized errors and other goodness-of-fit indices, than other methods [104,106]. Hu, Bentler and Kano (1992) state that these robust statistics (i.e., the Satorra–Benter scaled statistic (S-Bχ^2^) and robust standardized errors) have been shown to perform better than uncorrected statistics where the assumption of multivariate normality is not supported [107]. Considering this, we performed the robust maximum likelihood method to estimate the structural equation model [104].

The measurement model in Table 2 was derived from the CFA, indicating a satisfactory level of fit for all goodness-of-fit indices. These findings confirm that the proposed measurement model fits the data well: = 1261.989; S–B = 994.587; df = 549; normed fit index (NFI) = 0.890; non-normed fit index (NNFI) = 0.939; comparative fit index (CFI) = 0.947; and root mean square error of approximation (RMSEA) = 0.047. Table 2 also presents Cronbach’s alpha values which estimate the reliability of multi-item scales: Personal COVID-19 Quarantine Policy (PQP: 0.878), Company’s COVID-19 Quarantine Policy (CQP: 0.911), Policy (PLY: 0.936), Attitude (ATT: 0.908), Positive Anticipated Emotion (PAE: 0.920), Positive Expectation of COVID-19 (PEC: 0.836), Indoor Leisure (IDL: 0.903), Outdoor Leisure (OTL: 0.890), Intra-bound Tourism (IBT:0.884), and Smart Tourism (SMT: 0.897). All of these values demonstrate acceptable reliability for each construct because all alpha coefficients are greater than 0.7 [108].

All average variance extracted (AVE) and composite reliability (CR) values for the multi-item scales are greater than the minimum criteria of 0.5 and 0.7, respectively [100], indicating a sufficient level of convergent validity for the measurement model. To check discriminant validity of the constructs, the study uses three methods. The first method, AVE, confirms the discriminant validity because the AVE of each construct is greater than the squared correlation coefficients for corresponding inter-construct comparisons [109]. Furthermore, the other two methods (confidence intervals and constrained models) both support the discriminant validity. For details on measurement items, please refer to Table A1 in Appendix A.

### 4.3. Hypothesis Testing

Figure 2 summarizes the estimated results of the proposed research model in this study. The results confirm that the proposed structural model fits the data well: S–B = 1042.930, df = 569, NFI = 0.885, NNFI = 0.938, CFI = 0.944, and RMSEA = 0.047. The explained variance of the endogenous constructs (Figure 2) is 53.0% for Positive Expectation of COVID-19 (PEC), 12.2% for Indoor Leisure (IDL), 37.7% for Outdoor Leisure (OTL), 72.7% for Intra-bound Tourism (IBT) and 45.0% Smart Tourism (SMT). In terms of hypothesis testing, Coping Strategies for COVID-19, Attitude and Positive Anticipated Emotion (PAE) have a positive effect on Positive Expectation of COVID-19 (PEC) (β_PQP→PEC_ = 0.131, t = 2.505, *p* < 0.05; β_CQP→PEC_ = 0.228, t = 4.177, < 0.001; β_PLY→PEC_ = 0.160, t = 3.443, *p* < 0.001; β_ATT→PEC_ = 0.230, t = 3.686, *p* < 0.001; β_PAE→PEC_ = 0.199, t = 2.821, *p* < 0.01). Positive Expectation of COVID-19 (PEC) has a significant effect on Behavioral Intention (β_PEC→IDL_ = 0.349, t = 6.360, *p* < 0.001; β_PEC→OTL_ = 0.614, t = 13.003, *p* < 0.001; β_PEC→IBT_ = 0.853, t = 36.804, *p* < 0.001; β_PEC→SMT_ = 0.671, t = 19.925, *p* < 0.001).

## 5. Discussion

The findings of this study suggest that effective coping strategies related to COVID-19 may lead to an increase in positive expectations related to the virus. This aligns with previous research indicating that effective coping strategies can improve tourists’ expectations. The specific coping strategies that were found to be effective in this study were not specified [27,110]. The results of this study showed that corporate COVID-19 quarantine policies had the greatest impact on positive expectations between the three response strategies examined: government COVID-19 quarantine policies, corporate COVID-19 quarantine policies, and individual COVID-19 quarantine policies. The government’s COVID-19 quarantine policy was found to have the second-highest impact on positive expectations, while the individual’s COVID-19 quarantine policy had the lowest impact. These findings suggest that companies have played a significant role in shaping the expectations of individuals during the COVID-19 pandemic, and that effective corporate quarantine policies can contribute to a more positive outlook among the general population. It is important for businesses to carefully consider and implement effective quarantine measures in order to mitigate the spread of the virus and support the overall health and well-being of their employees and communities. Attitude has the strongest impact on positive expectations related to other coping strategies with emotion-focused coping. This is consistent with previous findings that the coping strategies of tourism destination companies as well as tourists’ attitudes have the stronger influence on positive expectations related to COVID-19 [10,14,34]. Prior research has reported that positive expectations were the most important determinants of behavioral intention [92,93,94]. The results of the present study show that positive expectations have the highest impact on intra-bound tourism, followed by smart tourism, with outdoor leisure and indoor leisure ranking lowest. These results represent that domestic tourists were likely to travel domestically, and that smart tourism, which can maintain social distancing under COVID-19, are impacted due to emotional factors rather than cognitive factors.

## 6. Implications

The COVID-19 pandemic has had a significant impact on the tourism industry, leading to a decline in the number of tourists and causing widespread economic disruption. In order to restore confidence and help businesses recover from this crisis, it is necessary to implement effective strategies that address the various challenges facing the industry. These strategies should be multi-faceted, competitive, and forward-thinking, and should involve all stakeholders in the tourism industry working together to address the negative impacts of the pandemic and pave the way for a more sustainable and successful future. In particular, it is important to focus on developing and implementing strategies that address the specific challenges and needs of events tourism, which has been particularly compromised by the pandemic. These strategies should aim to demonstrate a commitment to addressing the negative externalities of tourism, and should be based on the latest research and best practices in order to ensure a smooth and successful recovery post-pandemic.

Technology has played a critical role in addressing the COVID-19 pandemic and in reopening the tourism industry and economy. Solutions such as mobility tracing apps, robotized AI touchless service delivery, digital health passports and identity controls, and technologies for social distancing and crowd control have been implemented to ensure health and safety. Additionally, big data and real-time decision making have been utilized to respond to the crisis. Humanoid robots have been used to deliver materials and disinfect public spaces, while technologies for detecting body temperature and providing safety and security have also been implemented. Technology is seen as a means of normalizing surveillance and collecting and analyzing personal data for fast decision making in the context of the COVID-19 pandemic.

Tourists and tourism management agencies are closely connected, so it is important for travel destination companies to prioritize implementing COVID-19 precautions and requirements. This can involve employing qualified employees and giving them the necessary authority, utilizing technology to make travel easier and more exciting, increasing online appointments and experiences, and considering the needs of tourists in order to formulate the corresponding measures. To ensure that tourists maintain social distancing at tourist destinations and hotels, tourism destinations can utilize science and technology such as artificial intelligence-driven robots. It is also important for the government to consider the impact of COVID-19 and provide effective supervision and management of the epidemic situation in tourist destinations. This includes disseminating positive information and news to enhance public confidence and courage. As the domestic epidemic tends to stabilize, domestic travel can be encouraged in order to restore the domestic economy. The local government can collect applicable data through big data to make the destination more activated, workable, and sustainable.

## 7. Conclusions

Unlike previous epidemics such as SARS, Ebola, and H1N1, COVID-19 remains the deadliest and longest-lasting epidemic in the world. The global shock of the COVID-19 pandemic continues, and it is difficult to guarantee its complete end in 2022. While this uncertainty has had an unprecedented impact on the travel market, the travel demand is recovering strongly in some countries. Effective coping strategies and travel expectations can quickly overcome the considerable difficulties associated with travel. Therefore, this study examines the relationship between coping strategies, positive expectations, and travel intentions by exploring individual and social responses to COVID-19 by conducting a large-scale survey of Korean tourists. The results show how people can develop effective coping strategies and a positive vision for the post-pandemic tourism recovery in the context of the pandemic. Despite the theoretical and practical contributions of the study, several limitations need to be considered. This study was conducted during COVID-19. Evidently, tourists’ decision-making processes and travel intentions have been and will be influenced by the current situation in Korea. Due to the time, resources, and social-distancing constraints, the data for this research were collected by an online survey from tourists in Korea, which may limit the generalizability of the findings. Future research can apply a face-to-face survey and interview tourists to examine the causal relationships between the variables. Moreover, further studies can launch investigations in other countries (e.g., Western countries). Since this study was conducted for the domestic tourism industry, caution is needed in generalizing the study results for the international tourism industry. When the COVID-19 pandemic subsides, a study examining the changes in the nature of or preference for overseas travel is also expected to give practical implications for the tourism industry.

## Figures and Tables

**Figure 1 ijerph-20-01424-f001:**
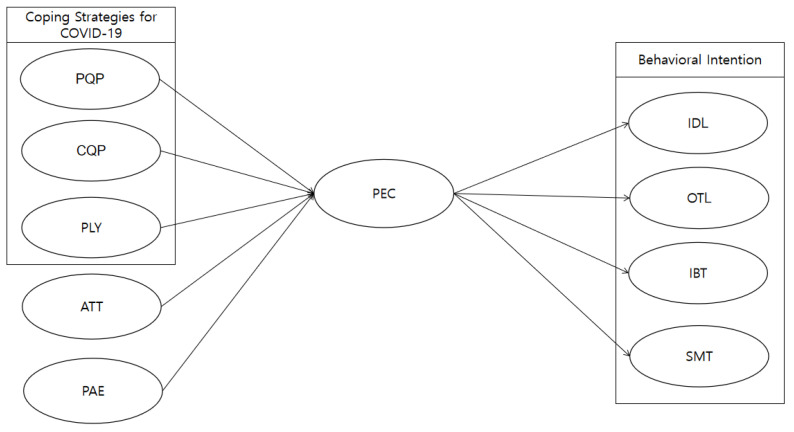
The proposed conceptual model. Note: ATT = Attitude; PAE = Positive Anticipated Emotion; PLY = Policy; PEC = Positive Expectation of COVID-19; BI = Behavioral Intention; PQP = Personal COVID-19 Quarantine Policy; CQP = Company’s COVID-19 Quarantine Policy; IDL = Indoor Leisure; OTL = Outdoor Leisure; IBT = Intra-bound Tourism; SMT = Smart Tourism.

**Figure 2 ijerph-20-01424-f002:**
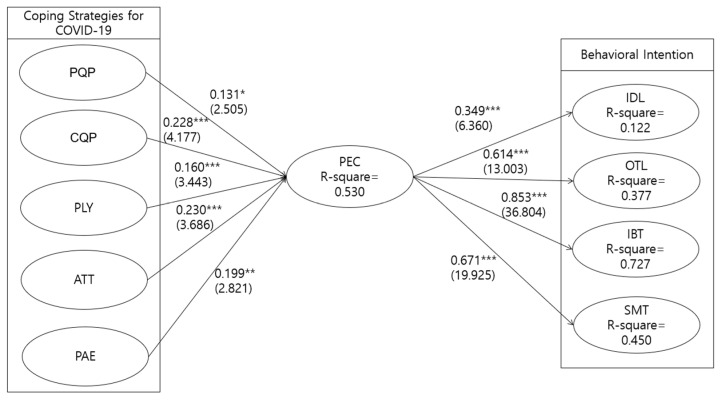
Structural model results. Note 1: * *p* < 0.05, ** *p* < 0.01, and *** *p* < 0.001. Note 2: Values not in parentheses are standardized parameter estimates; values in parentheses are t values. Note 3: ATT = Attitude; PAE = Positive Anticipated Emotion; PLY = Policy; PEC = Positive Expectation of COVID-19; BI = Behavioral Intention; PQP = Personal COVID-19 Quarantine Policy; CQP = Company’s COVID-19 Quarantine Policy; IDL = Indoor Leisure; OTL = Outdoor Leisure; IBT = Intra-bound Tourism; SMT = Smart Tourism. Note 4: S-Bχ^2^ (df) = 1042.930 (569), Normed S-Bχ^2^ = 1.833, CFI = 0.944, NFI = 0.885, NNFI = 0.938, and RMSEA = 0.047.

**Table 1 ijerph-20-01424-t001:** Respondents’ demographic characteristics (*n* = 375).

Characteristic	N (%)	Characteristic	N (%)
Gender		Marital status	
Male	194 (51.73)	Single	149 (39.73)
Female	181 (48.27)	Married	220 (58.67)
		Others	6 (1.60)
Education level		Monthly income level ^a^	
Less than high school	56 (14.93)	Less than KRW 1 million	50 (13.33)
Two-year college	59 (15.13)	KRW 1–1.9 million	38 (10.13)
Four-year university	218 (58.13)	KRW 2–2.9 million	103 (27.47)
More than university	42 (11.20)	KRW 3–3.9 million	68 (18.13)
		KRW 4–4.9 million	49 (13.07)
		KRW 5–5.9 million	36 (9.60)
		KRW 6–6.9 million	12 (3.20)
		KRW 7–7.9 million	9 (2.40)
		Over KRW 8 million	10 (2.67)
Occupation		Age	
Experts/Professionals	48 (12.80)	20–29	66 (17.60)
Businessmen/Self-employed	21 (5.60)	30–39	91 (24.27)
Service workers	23 (6.13)	40–49	114 (30.40)
Office workers	155 (41.33)	50–59	104 (27.73)
Official	17 (4.53)		
Students	29 (7.73)		
Housewives	48 (12.80)		
Freelancers	22 (5.87)		
Retire	1 (0.27)		
Others	11 (2.93)		

^a^: US $1 is equivalent to 1200 Korean Won.

**Table 2 ijerph-20-01424-t002:** The results of the measurement model.

Constructs	PQP	CQP	PIY	ATT	PAE	PEC	IDL	OTL	IBT	SMT	Items	Standardized Factor Loading
Personal COVID-19 Quarantine Policy (PQP)	**0.644**	0.216 (0.465)	0.141 (0.371)	0.179 (0.424)	0.282 (0.531)	0.223 (0.472)	0.066 (0.258)	0.160 (0.400)	0.225 (0.474)	0.137 (0.370))	PQP 1 PQP 2 PQP 3 PQP 4	0.824 0.775 0.822 0.788
Company’s COVID-19 Quarantine Policy (CQP)	0.032	**0.721**	0.267 (0.517)	0.166 (0.408)	0.203 (0.450)	0.268 (0.518)	0.077 (0.278)	0.282 (0.531)	0.303 (0.550)	0.140 (0.374)	CQP 1 CQP 2 CQP 3 CQP 4	0.802 0.866 0.848 0.877
Policy (PIY)	0.025	0.026	**0.786**	0.172 (0.414)	0.156 (0.395)	0.248 (0.498)	0.055 (0.234)	0.131 (0.362)	0.190 (0.436)	0.093 (0.304)	PIY 1 PIY 2 PIY 3 PIY 4	0.873 0.888 0.890 0.894
Attitude (ATT)	0.029	0.023	0.030	**0.769**	0.559 (0.748)	0.335 (0.579)	0.072 (0.269)	0.177 (0.421)	0.326 (0.571)	0.101 (0.318)	ATT 1 ATT 2 ATT 3	0.901 0.905 0.823
Positive Anticipated Emotion (PAE)	0.023	0.020	0.025	0.034	**0.796**	0.359 (0.599)	0.066 (0.258)	0.159 (0.399)	0.306 (0.553)	0.119 (0.346)	PAE 1 PAE 2 PAE 3	0.915 0.898 0.863
Positive expectation of COVID-19 (PEC)	0.021	0.022	0.024	0.026	0.024	**0.569**	0104 (0.322)	0.336 (0.580)	0.689 (0.830)	0.460 (0.678)	PEC 1 PEC 2 PEC 3 PEC 4	0.759 0.712 0.718 0.822
Indoor Leisure (IDL)	0.026	0.020	0.027	0.029	0.024	0.025	**0.759**	0.333 (0.577)	0.130 (0.361)	0.195 (0.442)	IDL 1 IDL 2 IDL 3	0.832 0.896 0.884
Outdoor Leisure (OTL)	0.026	0.024	0.027	0.028	0.024	0.027	0.030	**0.732**	0.500(0.707)	0.311(0.557)	OTL 1 OTL 2 OTL 3	0.824 0.884 0.858
Intra-bound Tourism (IBT)	0.027	0.027	0.029	0.034	0.029	0.031	0.030	0.034	**0.721**	0.407(0.638)	IBT 1 IBT 2 IBT 3	0.832 0.857 0.858
Smart Tourism (SMT)	0.021	0.021	0.023	0.024	0.022	0.029	0.024	0.027	0.033	**0.646**	SMT 1 SMT 2 SMT 3 SMT 4 SMT 5	0.625 0.681 0.867 0.913 0.891
CR	0.879	0.912	0.936	0.909	0.921	0.840	0.904	0.891	0.886	0.899	**Model fit** S-B χ^2^(df): 739.268 (467) Normed S-B χ^2^: 1.583 CFI: 0.975 NFI: 0.936 NNFI: 0.972 RMSEA: 0.034	
Cronbach alpha	0.878	0.911	0.936	0.908	0.920	0.836	0.903	0.890	0.884	0.897		
Mean	3.987	3.725	3.895	3.797	3.968	3.614	3.282	3.485	0.617	0.403		
SD	0.664	0.645	0.743	0.728	0.649	0.597	0.738	0.702	0.689	0.673		

Note: The values of AVE are along the diagonal and highlighted. Squared correlations among latent constructs are above the diagonal. Correlations among latent constructs are within parentheses. Standard errors among latent constructs are below the diagonal. Mardia’s normalized coefficient is: 71.445. All standardized factor loadings are significant at *p* < 0.001.

## Data Availability

The dataset used in this research are available upon request from the corresponding author. The data are not publicly available due to restrictions i.e., privacy or ethical.

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
