# Peer review of "A Study on the Structural Relationships between COVID-19 Coping Strategies, Positive Expectations, and the Behavioral Intentions of Various Tourism-Related Behaviors"

_ijerph, 2023, doi:10.3390/ijerph20021424_

Round 1

Reviewer 1 Report

The topic is very interesting and the model developed in the paper is innovative and inspiring. Comprehensive literature review kaes the paper valuable and reflects on scientific and intellectual curiosity of the Authors. There are shown implications of the research and the Authors are conscious of limitations and research gaps.

I don't any objections regarding the quality and usefulness of the article; congratulations to Authors 

Author Response

Thanks for your support. We will work harder to make more meaningful research.

Reviewer 2 Report

1. In the abstract, one sentence about the scientific novelty of the research should be added.

2. In the section "Methodology", it would be worthwhile to add a logical scheme of the research.

3. Authors should carefully check that all sources are referenced.

4. In the section "Discussion", authors should compare own results with existing ones in more detail, demonstrate their novelty, commonality and/or difference.

5. The results of the study presented in the reviewed article are interesting and valuable not only for China, but also for other countries, including Ukraine. This could be mentioned in the article, for example, in the section "discussion" and it is worth adding a few links to recent publications of Ukrainian scientists on this issue. In particular, the following sources should be added to the discussion:

Nazarova, K., Nezhyva, M., Moyseyenko, O., Mysiuk, V., Levkov, K., & Kucher, A. (2022). Tourism risk audit under the Covid-19 impact. Financial and Credit Activity Problems of Theory and Practice, 2(43), 53–62. https://doi.org/10.55643/fcaptp.2.43.2022.3608

Khudaverdiyeva, V., Merchanskyi, V., Chuiko, N., & Voronkova, A. (2022). Strategy of economic adaptation of the tourist-and-recreational sphere of Ukraine to the international tourist market. Journal of Innovations and Sustainability, 6(2), 02. https://doi.org/10.51599/is.2022.06.02.02

Kondratska, L. (2022). Development of the hospitality industry during the pandemic of COVID-19. Journal of Innovations and Sustainability, 6(3), 03. https://doi.org/10.51599/is.2022.06.03.03

6.    There should be no citations to sources in the conclusion. At the same time, in the conclusions, it should be noted in more detail about the results of testing the put forward hypotheses.

7. On page 14 it says "Appendix B", but there is no appendix.

Author Response

Thank you for your suggestion. We will work harder to make more meaningful research.

Reviewer 3 Report

Dear Authors,

The manuscript ‘A Study on the Structural Relationship between Corona-19 Coping Strategies, Positive Expectations, and the Behavioral Intentions of Various Tourism-related Behaviors’ discusses an important problem, presenting conclusions based on well selected method and well discussed findings and discussion. The data is sufficient to draw conclusions and the calculation method is described clearly enough to reproduce it.

Minor language mistakes appear, e.g. the sentence: ‘They are selected as best-fit participants by response to a sampling questionnaire and rejected during the survey if they complete their questionnaire too rapidly.’ should be put in simple past tense, to be consistent with the previous and following parts of the text.

The contribution of the paper is significant, and it has a potential to attract a wider international readership. That is why I recommend the paper to be published in its present form.

Author Response

(The authors gave the same response as above.)

Reviewer 4 Report

Abstract - well written although with repeating the long statements, please check and rewrite, redundancy of sentences/parts 

Check the language / correctly written sentences

Extensive and well written literature review

Reorganise the structure of the paper - the titles of the chapters (Theoretical framework - Literature review - merge and reorganise) 

The REC - Positive expectation of Covid - intriguing but not well explained major topic of the research. Please provide better explanation with citation (provide literature review of the concept of more authors). The name of the concept could be considered controversial. I explored it in WoS and Scopus database and found the citation only in one paper of author Song (possible self-citation).  Please provide better explanation.

Becaus of this aspect the major revision is suggested.

Methodology and results presentation are correctly written

Author Response

(The authors gave the same response as above.)

Reviewer 5 Report

The topic of the paper is current and interesting. Therefore, the results of this research are valuable. It would be interesting to conduct the research after the end of the COVID-19 pandemic and compare whether the results would be different compared to these.

There is an error in the numbering of the subtitles in the paper (the theoretical framework and methodology are numbered the same).

Author Response

(The authors gave the same response as above.)

Round 2

Reviewer 4 Report

The structure of the paper is problematic.

Author Response

Response to Reviewer Comments

            We sincerely appreciate the reviewers’ comments and suggestions on the previous version of this manuscript. We have thoroughly studied them and have revised the manuscript accordingly. This report summarizes our responses to all the comments (in red for your convenience).

  1. The structure of the paper is problematic.

Response: Thank you for your suggestion. We have checked and reorganized the structure of the paper.